# Peak Tibiofemoral Contact Forces Estimated Using IMU-Based Approaches Are Not Significantly Different from Motion Capture-Based Estimations in Patients with Knee Osteoarthritis

**DOI:** 10.3390/s23094484

**Published:** 2023-05-04

**Authors:** Giacomo Di Raimondo, Miel Willems, Bryce Adrian Killen, Sara Havashinezhadian, Katia Turcot, Benedicte Vanwanseele, Ilse Jonkers

**Affiliations:** 1Department of Movement Sciences, Katholieke Universiteit Leuven, 3001 Heverlee, Belgium; miel.willems@kuleuven.be (M.W.); bryce.killen@kuleuven.be (B.A.K.); benedicte.vanwanseele@kuleuven.be (B.V.); ilse.jonkers@kuleuven.be (I.J.); 2Department of Kinesiology, Université Laval, Québec, QC G1V 0A6, Canada; sara.havashinezhadian.1@ulaval.ca (S.H.); katia.turcot@kin.ulaval.ca (K.T.)

**Keywords:** knee osteoarthritis, knee contact forces, wearable sensors, IMU, musculoskeletal modelling, OpenSim, ground reaction forces, principal component analysis, joint moments

## Abstract

Altered tibiofemoral contact forces represent a risk factor for osteoarthritis onset and progression, making optimization of the knee force distribution a target of treatment strategies. Musculoskeletal model-based simulations are a state-of-the-art method to estimate joint contact forces, but they typically require laboratory-based input and skilled operators. To overcome these limitations, ambulatory methods, relying on inertial measurement units, have been proposed to estimated ground reaction forces and, consequently, knee contact forces out-of-the-lab. This study proposes the use of a full inertial-capture-based musculoskeletal modelling workflow with an underlying probabilistic principal component analysis model trained on 1787 gait cycles in patients with knee osteoarthritis. As validation, five patients with knee osteoarthritis were instrumented with 17 inertial measurement units and 76 opto-reflective markers. Participants performed multiple overground walking trials while motion and inertial capture methods were synchronously recorded. Moderate to strong correlations were found for the inertial capture-based knee contact forces compared to motion capture with root mean square error between 0.15 and 0.40 of body weight. The results show that our workflow can inform and potentially assist clinical practitioners to monitor knee joint loading in physical therapy sessions and eventually assess long-term therapeutic effects in a clinical context.

## 1. Introduction

Osteoarthritis (OA) is the most common chronic joint disease that affects the whole knee joint. Cartilage degeneration is the hallmark of OA, but associated changes in the subchondral bone, synovium, bone marrow, muscles, and ligament occur [1,2]. To date, OA is the leading cause of disability among the elderly, resulting in pain, limited daily activities, and a decreased quality of life [3,4,5]. Current treatments for patients with knee OA (PwKOA) are typically restricted to symptom relief and, in advanced stages of the disease, joint replacement surgery. To date, no known cure or proven strategy exists for stopping or slowing OA progression. This leads to a high and only increasing economic and societal burden of OA for patients and governments. In the US, annual direct and indirect costs attributable to arthritis and other rheumatic conditions are over $80 billion and $40 billion, respectively [1,6]. As such, methods for slowing the progression of the disease are a currently unmet clinical need with clear societal and economic benefits. To achieve these benefits, an understanding of the drivers of OA progression is required.

Altered joint loading (i.e., knee contact forces)—associated with obesity, malalignment, trauma, or joint instability—is a critical and known risk factor for the onset and progression of OA. Therefore, optimizing joint loading has been the target of conservative (e.g., orthosis, braces, and gait retraining) and surgical treatment strategies (e.g., high tibial osteotomy) in patients with knee OA [7,8]. Moreover, several studies have demonstrated that gait retraining—i.e., use of specific gait pattern modifications instructed by the physical therapist—can optimize knee joint loading, resulting in intermediate pain alleviation [9,10,11]. Therefore, ambulatory monitoring of joint loading parameters, i.e., as knee contact forces (KCF), has the potential to significantly impact disease management and rehabilitation of patients with knee osteoarthritis.

Direct measurement of knee contact forces—*in vivo*—requires invasive measurement techniques via instrumented knee implants that measure the forces transmitted through implants [12]. However, this approach is costly, invasive, and not feasible in large cohorts. Musculoskeletal (MSK) model-based simulations based on 3D motion capture (MoCap) data represent the state-of-the-art method to estimate in vivo joint contact forces. Using these methods, several groups have reported good agreement between MSK model-based KCF estimates and instrumented knee implant measurements [13,14].

Despite the accuracy, reliability, and repeatability of MSK model-based simulation workflows using laboratory-based MoCap, this approach is limiting true clinical implementation as it requires an expensive, high-tech, and controlled lab-based environment and relies on highly skilled operators [15]. In addition, patients may alter their movement pattern when walking in a highly controlled environment. As such, movement captured in these environments may not represent the natural walking pattern of patients in a “real” environment. To overcome this limitation, several ambulatory methods, relying on inertial measurement units (IMUs), have been developed [16,17,18]. Inertial capture systems (InCap) represent a valid alternative for camera-based motion capture systems when estimating joint kinematics following functional sensor-calibration [19,20,21,22,23]. Several studies have used InCap methods to also estimate kinetic parameters (i.e., joint moments) during daily living activities (stair ascent, descent, and sit-to-stand) based on machine learning approaches [24,25,26,27]. So far, these approaches were limited to estimating joint kinematics and moments, thereby neglecting the contribution of muscle forces. Only a few studies [28,29] used IMU-based data as input to a MSK-model- based simulation workflow to estimate joint loading profiles in patients with knee and hip OA. The results showed that to accurately estimate joint loading, the ground reaction forces are mandatory in addition to the IMU data. Therefore, to evolve to a real-world estimation of knee joint loading, a mobile estimate of ground reaction forces or even direct estimation of the joint moments from IMU data, which are then used as input to the musculoskeletal modelling workflow, is necessary.

Previous work from our research group [30] proposed estimating ground reaction forces and ground reaction moments (GRFM) during treadmill walking based on a probabilistic principal component analysis (PPCA) trained using gold standard MoCap data. Leave-one-out validation confirmed that this method works well for gait in healthy participants. However, the robustness of the workflow to capture gait modifications in subjects with KOA was not yet tested. Furthermore, the point of application of the estimated ground reaction forces (i.e., center of pressure), a necessary component for MSK modelling, was not estimated. The combination of the PPCA and zero moment point (ZMP) method [31,32,33], a previously validated method to estimate the center of pressure (COP), could potentially provide the missing input for a full IMU-based MSK model-based simulation workflow to estimate knee joint loading in a clinical context.

Therefore, the aim of this study is to develop a full InCap-based MSK modeling workflow for estimating knee joint contact forces (KCF), which we hypothesize will be sufficiently accurate to detect previously reported significant peak differences in KCF between healthy subjects and patients with knee OA in the order of 0.3 BW [34,35,36,37]. Such a workflow may be able to inform and assist clinical practitioners about the success of inducing knee joint loading modifications as part of the regular physical therapy sessions in an everyday life context, and eventually monitor the impact of feedback on an individual patient’s joint loading locomotor function during the disease progression.

## 2. Materials and Methods

### 2.1. Experimental Dataset Used to Develop and Validate the InCap Workflow

Training dataset: 3D MoCap collected at Laval University, Quebec City, Canada was used as gold standard reference dataset. Eighteen patients diagnosed with knee OA (demographic: 5 males and 13 females; age: 66.6 ± 7.3 year; height: 1.65 ± 0.12 m; weight 80.5 ± 15.3 kg; BMI: 29.86 ± 6.28 kg/m^2^, Kellgren-Lawrence score 2–4) walked on an instrumented treadmill at a self-selected speed (0.75 ± 0.23 m/s) for ~2 min (between 65 and 167 gait cycles per subject—1787 cycles in total). The 3D position of the 74 reflective markers, 42 attached to anatomical landmarks of the different body segments (ISB [38]) and 32 on clusters, was recorded using the 9 infrared camera system (VICON, Oxford Metrics Group, Oxford, UK, 100 Hz) while an instrumented treadmill (Bertec, Columbus, OH, USA, 1000 Hz) synchronously collected force plate data. All participants provided written informed consent prior to data collection. This research was in accordance with the ethical guidelines provided by the Ethical Research Committee Centre Intégré Universitaire de Santé et de Services Sociaux de la Capitale-National, Quebec (MP-13-2020-1954).

Validation dataset: 3D MoCap data was collected at the Movement & Posture Analysis Laboratory Leuven at the Department of Movement Science, KU Leuven, Leuven, Belgium, as the gold standard reference data system. Five patients with knee OA (demographic: 2 females and 3 males; age: 68.0 ± 3.7 year; height: 1.74 ± 0.7 m; weight 73.0 ± 12.11 kg; BMI: 23.95 ± 2.79 kg/m^2^, Kellgren-Lawrence score 3–4) walked overground at self-selected speed (1.16 ± 0.13 m/s) (between three and four gait cycles per subject—16 cycles in total). A total of 13 infrared camera systems (VICON, Oxford Metrics Group, Oxford, UK, 100 Hz) recorded the 3D position of 65 reflective markers attached on anatomical landmarks of the different body segments (Plug-in Gait [38]) as well as the ground reaction forces recorded using force plate (AMTI, Watertown, MA, USA, 1000 Hz). Simultaneously, a 3D InCap system (MVN BIOMECH Awinda, Xsens Technologies, Enschede, The Netherlands, 60 Hz [39,40]), consisting of 17 IMUs, recorded the 3D body segment orientation after performing a previously developed functional calibration procedure [41]. Both systems were time synchronized based on the manufacturer’s guidelines with a specific trigger at the start/stop recording time. All participants provided written informed consent prior to data collection. This research was approved by the Ethics Committees of the University Hospital Leuven in collaboration with Ziekenhuis Oost-Limburg (Genk) and Jessa Hospital (Hasselt, Belgium) (S59857).

### 2.2. Data Processing

Filtered, labelled, and gap-filled MoCap data were exported as .c3d files from Nexus 2.12 (training dataset) and raw InCap data were exported as .mvnx files from Xsens MVN 2021.0 into the InCap-based workflow (validation dataset) (Figure 1) developed in MATLAB (R2020a, MathWorks). All trials were processed and time-normalized to 100% of the gait cycle (101 time points) using custom-built MATLAB scripts.

### 2.3. InCap-Based Workflow Overview

The InCap-based workflow consists of four steps, schematically presented in Figure 1 and further elaborated below:

(1) The InCap validation dataset was used to calculate the joint kinematics following a previously developed functional calibration method—OpenIMUs [41]. (2) IMU-based kinematics were input to the probabilistic principal component analysis (PPCA) model to estimate the GRFM. (3) GRFM were used as input to the ZMP method to estimate COP position. (4) IMU-based kinematics and estimated FP data (GRFM and COP) were used in OpenSim Joint Articular Mechanics (JAM) [42,43] to estimate KCF. As validation, estimated KCF based on the MoCap data and InCap data were compared.

### 2.4. Musculoskeletal Model

A validated multi-body knee model with combined 12 degrees of freedom for the tibiofemoral (6DOF) and patellofemoral (6DOF) joints [35,43,44] was used. Fourteen ligaments surrounding the knee joint were represented by bundles of nonlinear elastic springs. Cartilage surface were modelled as a non-linear elastic material, and contact pressures were computed using an elastic foundation formulation [42].

The knee model was integrated into an existing lower extremity musculoskeletal model, which included 44 musculotendon units crossing the hip, knee, and ankle joints [45]. Cartilage contact pressures were calculated using a nonlinear elastic foundation formulation based on the penetration depth between overlapping vertices of the cartilage surface meshes. Combined uniformly distributed thicknesses of 3 mm were assumed in the tibiofemoral and patellofemoral joints, respectively [42,44]. An elastic modulus of 10 MPa and a Poisson ratio of 0.45 were assumed for cartilage [43]. The MSK model was scaled to subject-specific segment lengths, as determined from the marker positions measured during a static calibration trial.

### 2.5. PPCA Model Training Phase

A previously developed PPCA method was further customized [30] for use with KOA patients. In the training phase, MoCap-based kinematics and measured GRFM of 1787 gait cycles of 18 PwKOA were used to calculate joint moments using an inverse dynamics approach in OpenSim, which determines the generalized forces (e.g., net forces and moments) at each joint underlying a given movement. Inverse dynamic results were used to build the PPCA model based on PwKOA during treadmill walking. The new trained PPCA model was used to estimate the GRFM using as input the kinematics, the scaled MSK models, and the timing events of the stance phase [46,47]. The timing events specified in the PPCA model are needed to overcome the indeterminacy of the double stance phase and to reduce the sensitivity to skeletal model inaccuracies. The PPCA model then estimates GRFM that were most consistent with the observations of the training dataset. The model was validated based on leave-one-out-subject cross-validation [24].

### 2.6. PPCA Model Validation Phase

In the validation phase, the customized KOA-based PPCA model was used to estimate the GRFM using as input the InCap-based kinematics, the scaled MSK models, and the timing events during the overground walking. The InCap-based kinematics were calculated using a previously validated functional calibration method using functional movements [41,48]. Models were scaled based on the body segment dimensions derived from markers on anatomic landmarks [49,50].

### 2.7. COP Estimation

Calculation of knee joint contact forces requires estimations of ground reaction forces, moments (from PPCA), and center of pressure. To estimate the COP position, a previously validated Zero Moment Point (ZMP) method was applied [32,33].

However, during the heel strike and the toe-off instants (0% and 100% of stance phase), the ground reaction forces are close to zero. In the ZMP method equations [33], the coordinates of the center of pressure are obtained by dividing ground reaction moments by the ground reaction forces, and, therefore, if the forces are close to zero, the results tend to be infinite, inducing the discontinuities. To avoid these discontinuities, the estimated coordinates between 0–15% and between 85–100% of the stance phase were fitted using a polynomial function of the 10th order that best represents the entire COP curve to eliminate discontinuities. In addition, the fitted COP trajectory was based on the MSK model calcaneus origin position between 0% and 15% and between 85% and 100% of the stance phase. The estimated COP position was then compared to the measured COP position obtained from the force plates for validation.

### 2.8. Knee Contact Forces and Joint Moments Estimation

OpenSim JAM [44,51] with the integrated concurrent optimization of muscle activations and kinematics (COMAK) algorithm [44] was used to solve the muscle activation distribution problem, and compute the resultant secondary degrees of freedom (dofs) and the knee contact forces [52]. Subsequently, an inverse dynamics approach computed the external joint moments: knee flexion moment (KFM), knee adduction moment (KAM), and knee rotation moment (KRM) for the three rotational knee dofs based on the resultant optimized kinematics. For the InCap-based knee contact forces estimation, IMU-kinematics (from OpenIMUs) and the estimated GRFM and COP position (from PPCA and ZMP) were used. In addition to knee contact forces, secondary knee coordinates and moments were also compared between both approaches (Appendix A).

### 2.9. Statistics

The Shapiro–Wilk test revealed that the KCF data were not normally distributed. Thus, for multiple comparisons, the Wilcoxon rank test was used to statistically compare KCF peak and impulse difference between MoCap- and InCap-based results in MATLAB. A significance level was set at *p* ≤ 0.05. A false discovery rate (FDR) statistical approach in multiple assumptions testing was used for multiple comparisons in order to correct for random events that falsely appear significant. A q-value threshold of 0.05 (FDR of 5%) among all significant variables was used. To evaluate the functional relevance of the observed differences between MoCap and InCap-based knee contact forces estimation, the root mean squared error (RMSE), the determination coefficient R^2^, and the mean absolute differences in first, second peak, and impulse were evaluated. Errors were compared to previously reported KCF differences between the healthy subject and PwKOA (in the order of 0.45–0.60 BW for the first KCF peak, 0.30–0.45 BW for the second KCF peak and 0.3–0.6 BWs for the KCF impulse [36,37,53,54]) in order to evaluate the feasibility of the frameworks to differentiate the two––a key goal for clinical applications.

## 3. Results

Complete figures and tables for a comparison of estimated and measured parameters are presented in the Appendix A.

**GRFM estimation (Figure 2)**: The customized PPCA model showed an average RMSE of the vertical, anterior-posterior, and medio-lateral ground reaction forces of 0.083 ± 0.035 BW, 0.020 ± 0.008 BW and 0.017 ± 0.096 BW, respectively, with R^2^ of 0.97 ± 0.26, 0.86 ± 0.14, and 0.85 ± 0.26, respectively; further, an average RMSE of the sagittal, frontal, and medio-lateral ground reaction moments of 0.198 ± 0.038 Nm/kg, 0.202 ± 0.066 Nm/kg and 0.041 ± 0.014 Nm/kg was shown, respectively, with R^2^ of 0.93 ± 0.21, 0.59 ± 0.09, and 0.66 ± 0.18, respectively.

**COP estimation (Figure 3):** The ZMP method showed an average RMSE of 1.26 ± 0.58 cm and RMSE of 0.66 ± 0.30 cm for the anterior-posterior position and the medio-lateral position, respectively, with R^2^ of 0.98 ± 0.11 and 0.88 ± 0.19, respectively.

**KCF estimation (Figure 4):** The estimated resultant IMU-based KCF showed an average RMSE of the medial and lateral knee compartment of 0.35 ± 0.11 BW and 0.15 ± 0.05 BW, respectively, and moderate to strong R^2^ of 0.76 ± 0.12 and 0.58 ± 0.24, respectively (Table 1). The mean absolute difference (MAD) for the first peak, second peak, and impulse of the medial knee compartment contact force was of 0.21 ± 0.14 BW, 0.27 ± 0.14 BW and 0.30 ± 0.19 BWs, respectively. Further the MAD for the first peak, second peak, and impulse of the lateral knee compartment contact force was of 0.08 ± 0.06 BW, 0.12 ± 0.13 BW and 0.31 ± 0.15 BWs, respectively (Table 2). Significant differences between MoCap and InCap-based estimates were only confirmed for total and medial KCF impulse (Appendix A).

## 4. Discussion

The aim of this study was to develop an InCap-based MSK modeling workflow for monitoring knee joint contact forces that is accurate enough to discriminate between healthy subjects and patients with KOA. The developed method addresses previously documented research gaps within the literature, specifically through the use of the probabilistic Bayesian principal component analysis modelling based estimation of ground reaction forces, ground reaction moments, and the consequent calculation of the COP position using an inertial capture system combined with a MSK modelling workflow without the requirement of lab-based force plates. In general, results showed that the developed workflow’s accuracy, in terms of peak knee contact forces and impulse differences (<0.27 BW and <0.31 BWs, respectively), does allow for the detection of previously reported differences in KCF between healthy people and patients with KOA (0.45–0.60 BW for the first KCF peak, 0.30–0.45 BW for the second KCF peak, and 0.30–0.60 BWs for the KCF impulse [36,37,53,54]).

The customized probabilistic principal component analysis (PPCA) model was trained on 18 patients with KOA, and it is the first novel step within this InCap-based MSK modelling framework that can facilitate the process. The accuracy of this approach was comparable to the previously developed PPCA model trained on 23 healthy adults [30], which were the vertical, anterior-posterior, and medio-lateral ground reaction forces RMSE of 0.050 ± 0.040 BW, 0.050 ± 0.040 BW, and 0.050 ± 0.040 BW, respectively, and the sagittal, frontal, and medio-lateral ground reaction moments RMSE of 0.098 ± 0.013 Nm/kg, 0.092 ± 0.001 Nm/kg, and 0.069 ± 0.020 Nm/kg, respectively. Previous methods for the estimation of GRFM described in the literature (e.g., smooth transition assumption (STA) [55], zero moment point method [33] and the optimization method [56]) reported errors in the magnitude of GRF > 0.08 BW and GRM > 0.15 Nm/kg. Our customized PPCA model presented similar accuracy, with errors in GRF < 0.083 BW and GRM < 0.202 Nm/kg. Only artificial neural network (ANN) methods [57] outperformed our PPCA model resulted in lower errors (GRF < 0.07 BW and GRM < 0.10 Nm/kg). However, it is only fair to comment that, when using the ANN method, parameter estimation is highly sensitive to and dependent on the input data. Hence, the validity of the ANN model for unseen situations, i.e., not used in initial training, is inherently limited. Therefore, it is important to emphasize that the reported performance of our PPCA model accounts for its generalizability given that, despite our model being trained on treadmill data, it was able to accurately predict untrained situations, in particular overground walking. Therefore, the developed PPCA model is generalizable for both treadmill and overground walking.

The embedded MSK-modeling workflow relies on an accurate estimation of the center of pressure using the ZMP method. Several studies described accurate methods to estimate the COP position based on machine learning approaches. Oubre et al. [58] estimated, with low-cost wearable devices and supervised machine learning models, anterior-posterior and medio-lateral COP with an average RMSE of less than 1.5 cm and 0.6 cm, respectively. Podobnik et al. [59] estimated the anterior-posterior and medio-lateral COP with an average RSME of 1.49 and 0.09 cm, respectively, solely from raw IMU data using a linear model and a non-linear Long-Short-Term Memory (LSTM) neural network model. Overall, the different methods for the COP estimation based on wearable sensors reported an average RMSE of less than 1.5 cm for the anteroposterior COP and less than 0.8 cm for the mediolateral COP [60,61], which is comparable to our customized ZMP method with an average RMSE of 1.26 cm for the anteroposterior COP direction and 0.66 cm for the mediolateral COP direction.

The overall accuracy of the developed workflow allows estimation of compartmental knee contact forces with an accuracy that allows discriminating knee loading conditions in patients with KOA from healthy controls (peak knee contact forces and impulse differences less than 0.27 BW and 0.31 BWs, respectively). Previous studies have also proposed wearable sensor-based methods to estimate KCF from measured GRFM combining IMUs with force-sensitive resistors (FSRs) in order to measure GRFM and COP position. However, the estimation of knee contact forces using these FSRs sensors is also challenging, as it requires the integration of external forces and moments over time, which can be affected by measurement noise and sensor drift [14,62,63]. Other methods include the use of pressure insoles, which can provide estimates of the normal force at the foot, and the use of machine learning algorithms to predict knee contact forces based on IMU data. A recent study by Stetter et al. [64] used an ANN to estimate knee contact forces based on IMU data, and it reported predicted vertical KCF peak and resultant vertical KCF across different movements (walking, running, and jumping) by an average of 0.17 ± 0.14 BW and 0.06 ± 0.06 BW, respectively. Related work by De Brabandere et al. [28] showed that a combination of one IMU sensor and machine learning can estimate knee joint loading profiles for an unseen patient, with a mean absolute error of 29% BW, although for more accurate knee contact forces estimates, both IMU-kinematic variables and ground reaction forces are needed. Other published methodologies for estimating knee contact forces based on MSK modelling, MoCap kinematics, and estimated GRFM reported an error around 0.3 BW [14,65]. Therefore, our developed method based on MSK modelling, InCap-kinematics and estimated GRFM provide an accurate and promising knee contact force estimation comparable to gold standard MoCap-based approaches.

The study presented has some limitations that need to be addressed in future research. The ZMP method showed strong discontinuities in the estimation of COP position during the initial and final double support phases. In future, its performance may be refined by adjusting its parameters and calculations using a set of measured data to be trained in the PPCA model to improve its accuracy. Another drawback of the study was that the PPCA model was tested on a small sample size consisting of only five PwKOA, with a total of 16 gait cycles. This might restrict the applicability of the workflow to a wider range of OA involvement. Moreover, further research should consider the use of the PPCA model for the estimation of knee contact forces, including other daily life activities such as stair negotiation.

Within the current context of use, the performance of the developed PPCA and InCap-based, MSK modeling workflow warrants the potential of estimating knee contact forces in an ecological environment during walking. Moreover, the developed approach demonstrated to be highly generalizable compared to previous pure AI-based approaches; due to the model scaling, the PPCA model can be used for subjects with a large difference in body mass, height, and gait speed (see demographic in Section 2). AI-based approaches rely on a high volume of training data, and their black box nature means new situations, gait patterns, and environments likely result in a large increase in the errors. Within the developed workflow, the use of appropriate IMU calibration methods seems crucial given that kinematic errors exceeding 5° will result in inconsistent estimation of COP from the estimated GRFM and induce high inaccuracies in KCF estimation. Therefore, it is essential to apply state-of-the-art calibration techniques, such as the one previously developed by our group, when validating the ground truth InCap-based results.

## 5. Conclusions

In conclusion, the developed mobile workflow based on musculoskeletal modelling, sensor-to-segment calibration (OpenIMUs) [41], the PPCA [30] model, and the ZMP [32] method offer a relatively simple and cost-effective approach to estimating knee contact forces, and they have the potential to be used in a variety of research and clinical contexts. Therefore, the developed workflow would eventually allow monitoring KCF in an ecological context and a consequent impact of specific gait interventions on an individual patient’s locomotor function and joint loading. This is of high clinical importance to inform clinical practitioners on how to induce joint loading changes as part of the regular therapy sessions with the aim to reduce activity-related pain and functional decline.

## Figures and Tables

**Figure 1 sensors-23-04484-f001:**
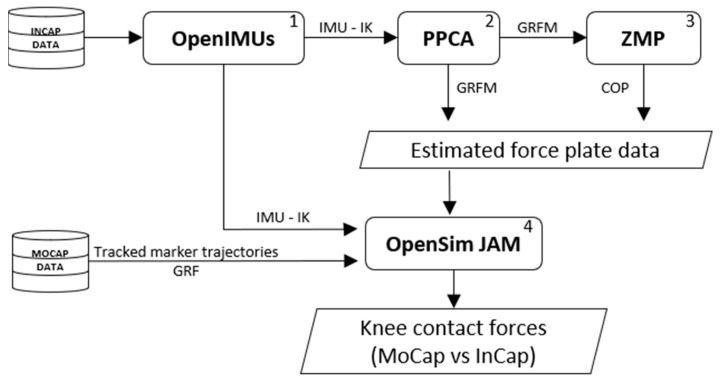
InCap-based workflow overview. (1) OpenIMUs—IMU calibration and inverse kinematics estimation; (2) PPCA model—estimation of GRFM; (3) ZMP—estimation of COP; (4) OpenSim JAM—KCF estimation and comparison.

**Figure 2 sensors-23-04484-f002:**
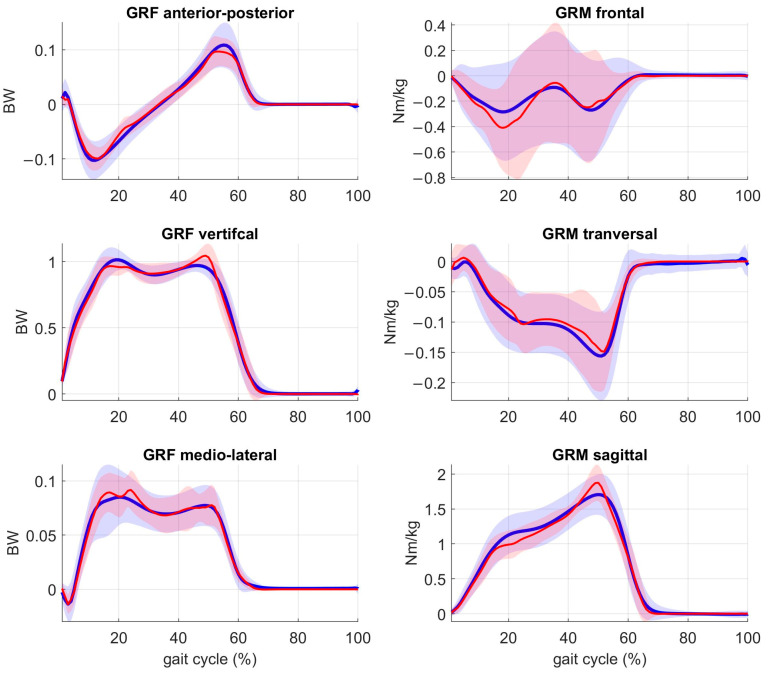
Measured (blue) vs. estimated (red) ground reaction forces (GRF) in BW (body weight) and ground reaction moments (GRM) in Nm/kg of the PPCA KOA population-based model (1787 gait cycles). Note that the illustrated comparison refers to the OA-involved knee joint. The shaded areas represent the standard deviation of the mean.

**Figure 3 sensors-23-04484-f003:**
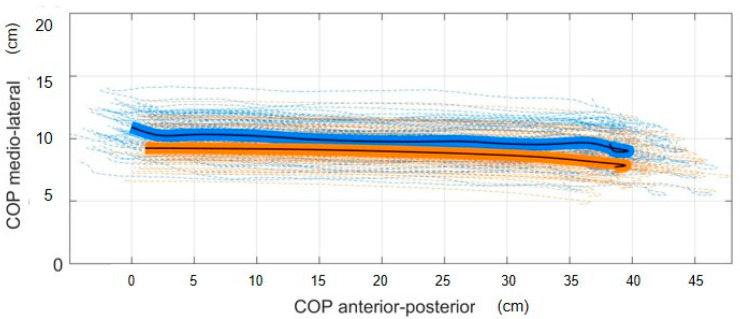
Mean measured (blue solid) vs. mean estimated (orange solid) COP trajectories of one patient during walking (84 gait cycles) (COP trajectories in dashed—measured in blue and estimated in orange).

**Figure 4 sensors-23-04484-f004:**
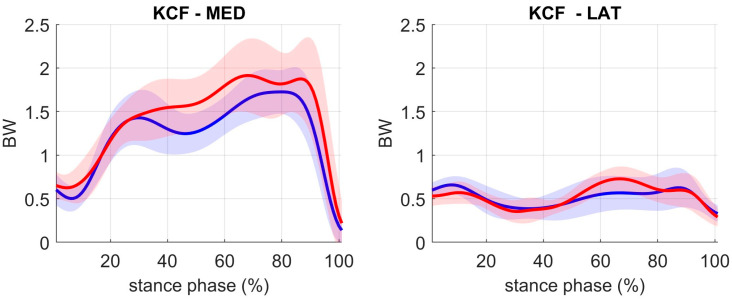
MoCap-based (blue) vs. InCap-based (red) knee contact forces in BW (body weight) for the medial and lateral knee compartment of the OA-involved limb. The shaded areas represent the standard deviation of the mean.

**Table 1 sensors-23-04484-t001:** Root mean square error (RMSE) and coefficient of determination (R^2^) of knee contact forces between MoCap and InCap.

Knee Contact Forces	RMSE (BW)		R^2^	
	Mean	Std	Mean	Std
Total	0.40	0.17	0.68	0.16
Medial	0.35	0.11	0.76	0.12
Lateral	0.15	0.05	0.58	0.24

**Table 2 sensors-23-04484-t002:** Mean absolute difference (MAD) in first peak, second peak, and impulse between MoCap and InCap estimated total knee contact forces, and in the medial and lateral knee compartment.

		MAD	
		Mean	Std
Peak 1 (BW)	Total	0.24	0.15
	Medial	0.21	0.14
	Lateral	0.08	0.06
Peak 2 (BW)	Total	0.19	0.15
	Medial	0.27	0.14
	Lateral	0.12	0.13
Impulse (BWs)	Total	0.31	0.18
	Medial	0.30	0.19
	Lateral	0.31	0.15

## Data Availability

The workflow and data will be made available on RDR—the KU Leuven repository.

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
