# Peer review of "Peak Tibiofemoral Contact Forces Estimated Using IMU-Based Approaches Are Not Significantly Different from Motion Capture-Based Estimations in Patients with Knee Osteoarthritis"

_sensors, 2023, doi:10.3390/s23094484_

Round 1

Reviewer 1 Report

The scope of the paper is to show that peak tibiofemoral contact forces in patients with knee osteoarthritis obtained from a motion capture system and involving the probabilistic principal component analysis correspond to those measured by inertial measurement units.

Areas of strength: Authors try to develop a new modeling workflow that can help to monitor knee joint loading in physical therapy sessions and to assess long-term therapeutic effects. There are 64 references (Have the authors checked if all of them are cited in the text)?

Weakness:

Are the raw data obtained from the motion capture system insufficient to aid the therapy of patients with knee osteoarthritis? Could the authors explain that their approach introduces a new quality? How can they prove that the errors in estimating contact forces are low in every ‘real’ everyday condition and do not deliver false information?

My remarks:

Page 1, line 25: longitudinal? - not long-term?
Page 1, line 40: costs are
Page 2, line 67: in ‘real’ everyday conditions – in my opinion, the word ‘ecological’ is not needed (also in line 342)
Page 2, line 92: in ‘real’ everyday conditions.
Page 3, line 98: what does ‘therapy in an ecological context’ mean?
Page 4, line 150: a typo ‘estiamtion’
Page 4, 156, and 159: Why is ‘cartilage surface contact pressure’ modeled as a linear elastic material and ‘cartilage contact pressure’ - as a nonlinear elastic material? What is the difference between them?
Page 5, line 190:  This paragraph needs to be clarified (discontinuities, polynomial of the 10th order). Did only one single coordinate represent the COP? How is it related to Fig. 3?
Page 5, line 226: How to relate the RMSE value of 0.083 with data shown in Fig. 2? What do shaded areas in Fig. 2 (Fig. 4) mean? Why are the results in Fig. 2 not compared with the literature data?
Page 7, line 271: What does ‘challenges within literature’ mean?
Page 8, line 277: ‘would allow’ – Does it work or not?
Page 8, line 293: I do not understand this sentence: “Therefore… walking”
Page 8, line 321: Stetter et al.  [?] – which study?
Page 8-9, lines 326-327: unclear from the word ‘but that …needed’
Page 9, line 331: Provide or not provide?
Page 9, line 334: Could you clearly explain these strong discontinuities? Is it surprising that COP moves during the gait cycle?
Page 9, 339: Consider rewriting: “the generalizability of the findings”
Page 9, 345: What is the source of these data?

Author Response

I attach the revisions and answer to your comments/feedback. 

Thanks 

Reviewer 2 Report

The paper seems interesting. The aim of this study is to develop a full Inertial Capture (InCap) based musculoskeletal (MSK) modeling workflow for estimating knee joint contact forces (KCF). The authors want to provide a simpler and cheaper method based on inertial measurement units as an alternative model for camera-based motion capture systems when estimating joint kinematics following functional sensor-calibration. The Introduction is adequate to explain to the reader the problem and what the authors proposed to solve it. Materials and methods are very well described. The results are well described and coincide whith what the authors proposed. the discussion is very well developed and complete and the conclusions are well supported by the results.

In conclusion, I think that the paper coincides with the scope of the journal and is suitable for publication in it.

Author Response

We would like to extend our gratitude to the reviewer for taking the time to provide positive feedback on our paper. We are delighted to hear that you found our work to be positive and impactful. Your encouragement motivates us to continue our efforts in producing high-quality research that contributes to the field. Thank you again for your support.